# Morphological changes in the human mandible associated with the presence of exostoses: A cross-sectional study in two archaeological populations from southern France

Estelle Casazza[1]*, Benoit Ballester[2], Camille Philip-Alliez[3], Yann Ardagna[4], Anne Raskin[1]

1 Aix Marseille Univ, APHM, CNRS, EFS, ADES, Hôpital de la Timone, Pôle PROMOD ODONTO, Service d'Odontologie Hospitalière et Chirurgie orale, Marseille, France, 2 Aix Marseille Univ, APHM, INSERM, IRD, SESSTIM, ISSPAM, Hôpital de la Timone, Pôle PROMOD ODONTO, Service de Réhabilitations Orales, Marseille, France, 3 Aix Marseille Univ, APHM, University Gustave Eiffel, LBA, Hôpital de la Timone, Pôle PROMOD ODONTO, Service d'Orthopédie Dento-Faciale et d'Odontologie Pédiatrique Générale, Marseille, France, 4 Aix Marseille Univ, CNRS, EFS, ADES, Marseille, France

* estelle.casazza@ap-hm.fr

**Data Availability Statement:** All relevant data are within the paper and its Supporting information files.

## Abstract

This cross-sectional study aimed to investigate morphological changes in the human mandible in archaeological collections associated with the presence of bony exostoses at the mandibular angle, which is described in the literature as related to specific behaviours of the masticatory apparatus like parafunctional activities. The skeletal sample investigated comprised adult individuals from two archaeological series. Sixteen measurements of the mandible were selected to evaluate anatomic variation in the two populations. Mandibles from both series were pooled for statistical analysis into two groups according to the number of exostoses at the mandibular angles: group 1 (number of exostoses $\leq 1$) and group 2 (number of exostoses $\geq 2$). Measurements could be made on eighty mandibles. A statistically significant difference was demonstrated between group 1 and group 2 for the following parameters: distance between mandibular angles, bicondylar width, ramus height, left condyle length, and right and left coronoid process height. For each of these parameters, values were higher in group 2 than in group 1. This study presents an original methodology for studying anatomical variations of the mandible in the context of parafunctional activity, highlighting certain mandibular modifications. The impact of parafunctional behaviours such as bruxism on the mandible therefore has many anatomical expressions. The considerable variability of results found in the literature shows that more studies are needed to reach a consensus on the impact of parafunctional activities on the mandible.

**Funding:** The author(s) received no specific funding for this work.

**Competing interests:** The authors have declared that no competing interests exist.

## Introduction

The mandible, the only mobile bone in the human skull, consists of three parts: the body (corpus) and two rami, connected by two mandibular angles. Like all skeletal bones, the mandible is constantly remodelled under the effect of several factors, which may be systemic (e.g. hormones, growth factors) or local (e.g. mechanical stresses) resulting from the action of the muscles of the masticatory apparatus [1–3]. Several muscles are inserted into the mandible to support the functions of mastication, swallowing, phonation, and emotion management. These include the mandibular elevator and depressor muscles, and those responsible for mandibular propulsion [4], which insert at various mandibular levels and connect the mandible to the facial skeleton. More specifically, for mastication, four main muscles are described: the masseter, temporalis, medial pterygoid, and lateral pterygoid muscles.

Bone remodelling induced by mechanical stress was described as early as 1892 by Wolff, who noted that "bone in a healthy person or animal adapts to the forces to which it is subjected" [5]. Bone can respond locally to the mechanical stresses to which it is subjected, via cellular adaptation mechanisms [6]. Several types of bone response have been described. If appropriate mechanical stresses are applied, bone mass may increase. However, if the mechanical stresses applied are too great, there may be a reduction in bone mass through increased resorption, with a risk of fatigue fracture. On the other hand, if the bone is not stressed, or is immobilized, it exhibits a high level of resorption with a concomitant reduction in bone formation [7].

Assessment of mandibular bone remodelling involves the evaluation of mandibular volume. Here, researchers have several options: the use of radiological imaging derived from medical activity, which requires the use of slices in all three spatial planes [8]; accessing bodies donated to science, which presents difficulties of access and implementation (e.g. preparation of parts); the study of archaeological series, which is dependent on the excavation conditions and state of preservation of the individuals recovered.

While the study of archaeological series deprives the researcher of access to certain data on the subjects investigated, it does allow measurements to be collected directly from the mandible, when its state of preservation permits. This has allowed the evaluation of mandibular markers, described in the literature as related to specific behaviours of the masticatory apparatus. This is the case for exostoses that may be present at the mandibular angles [9]. The location of these bony protuberances corresponds to the lower insertion zones of the masseter muscle, the most powerful elevator muscle in the masticatory apparatus, and allows mastication [10] Indeed, masseter contraction produces forceful elevation of the mandible to close the mouth. It also contributes to mandibular propulsion and biting. These exostoses develop when repeated muscular forces are applied through frequent contraction of the masseter muscles; the mandibular angles can also tend to rotate, angulate or bow themselves because of the extensive forces [11]. Several exostoses may be found at the same angle. The number of these exostoses reflects both the intensity of the forces exerted on the mandible in this area, and the high frequency with which these forces occur in the life of the individual, exceeding those ordinarily applied for masticatory functions alone, and potentially reflecting parafunctional behaviours established in the masticatory apparatus of the individuals observed [12].

This study aimed to investigate morphological changes in the mandible of human subjects in archaeological collections associated with the presence of bony exostoses at the mandibular angle.

## Materials and methods

This study followed the recommendations of the STROBE Statement checklist (see S1 Appendix).

## Skeletal samples

This study was conducted with ethical approval from the SRA DRAC-PACA (Services Régionaux de l'Archéologie; Direction Régionale des Affaires Culturelles–Provence Alpes Côte d'Azur), French Ministry of Culture. No permits were required for the described study, which complied with all relevant regulations. The skeletal sample investigated comprised adult individuals from two archaeological series, Les Petites Crottes (Bouches-du-Rhône, Marseille, Old cemetery of Crottes, 2016, 10 461), and L'îlot Saint Jacques (Bouches-du-Rhône, La Ciotat, Carré Saint Jacques, 2011). Both collections are held at the SRA DRAC PACA Osteotheque (Marseille, France) and were studied on-site.

**Description of archaeological sites.** The Petites Crottes sample, dated from 1784 to 1905, was excavated from a parish cemetery in North Marseille, in southern France. Excavations took place in 2013 and 2014 [13]. The field study population consisted of 1,548 individuals; 532 subjects were studied in the laboratory [14]. Adult sex was evaluated in most cases by an assessment of the coxal bone using Bruzek's morphoscopic method [15] and probabilistic sex diagnosis or PSD [16]. Age at death of adult subjects was estimated by noting the bone maturity status of the clavicle and the coxal bone, degenerative pathologies, and dental status (attrition, ante-mortem tooth loss, etc.). Adult subjects were divided into three age classes: young adults (20–39 years), mature adults (40–59 years), and elderly adults (> 60 years).

The Ilot Saint-Jacques sample came from a parish cemetery, in use from 1581 to 1831, in the town of La Ciotat, in southern France. Excavations took place in 2009 [17]. The population studied in the field comprised 1,245 individuals; 400 subjects were studied in the laboratory. Biological parameters were investigated using techniques identical to those described for the first series.

**Sample selection.** The first phase of selection of individuals from the subjects preserved in the SRA DRAC-PACA Osteotheque in January 2023 was based on the individual osteological conservation sheets taken from the excavation reports for the two archaeological sites, according to the following criteria:

Subject inclusion criteria

- Adult individual,

- With or without a sex diagnosis,

- Presenting a completely intact mandible OR presenting a mandible with a single post-mortem bone fracture allowing anatomical reassembly of the two fragments present OR an incomplete mandible missing a single mandibular anatomical element such as a condyle, coronoid process, or mandibular angle (right or left).

Criteria for non-inclusion of subjects

- Immature subject,

- Presence of temporary teeth on an arch,

- Mandible absent,

- Mandible deterioration: in more than two pieces, several fragments missing,

- Edentulous mandible (ante-mortem closure of dental sockets).

In the second phase (from March to December 2023) the collections were examined, and the subjects listed in the initial excavation report file as meeting the inclusion criteria but presenting significant deterioration of the mandible objectified during the examination were excluded.

## Methods

### Measurements made on archaeological series

Some of the measurements made were based on previously defined procedures for dry bone, except for condylar and coronoid process measurements [18–20]. Sixteen measurements of the mandible were selected to evaluate anatomic variation in the two populations:

- Chin height, right side: the distance from infradentale (the highest and most anterior point on the alveolar process in the median plane between the mandibular central incisors) to gnathion (point on the inferior mandibular border in the midsagittal plane),

- Corpus height, right side: the distance from the alveolar process to the inferior border of the mandible at the level of the mental foramen,

- Corpus width, right side: the maximum breadth measured at the level of the mental foramen perpendicular to the long axis of the mandibular body,

- Bigonial breadth: the distance between the right and left gonion (point on the mandibular angle where the outline of the mandible intersects with the line bisecting the angle formed by the line tangent to the posterior ramus border and the line tangent to the inferior border of the body),

- Bicondylar breadth: the distance between the most lateral points on the mandibular condyles,

- Minimum ramus width, right side: the minimum breadth of the mandibular ramus measured perpendicular to the height of the ramus,

- Maximum ramus height, right side: the distance from the gonion to the highest point on the mandibular condyle,

- Corpus length, right side: the distance from the anterior margin of the chin to the midpoint of a straight line extending from the posterior border of the right and left mandibular angles,

- Mandibular angle, both sides: the angle formed by the inferior border of the corpus and the posterior border of the ramus,

- Maximum width of right and left condyles: the distance between the most lateral points on each mandibular condyle,

- Maximum length of right and left condyles: the distance between the two furthest points in the width direction perpendicular to the long axis of each condyle,

- Presence and number of exostoses at the mandibular angles,

- Height of right and left coronoid processes: the distance between the mandibular notch and the coronoid process.

Measurements and their definitions are indicated in Fig 1a–1d.

These measurements were performed independently by two calibrated operators (EC, BB) from all available skulls meeting the inclusion criteria. Measurements were recorded using sliding calipers (Dexter, Lille, France), with readings being taken to the nearest 0.01 mm, while a mandibulometer (Siber Hegner, Lyon, France) was used to measure the mandibular angle, the maximum ramus height and the corpus length. Not all measurements were recorded on every individual due to the incomplete or fragmentary condition of the remains. Nevertheless, most measurements could be recorded on all individuals. Measurements recorded on parts

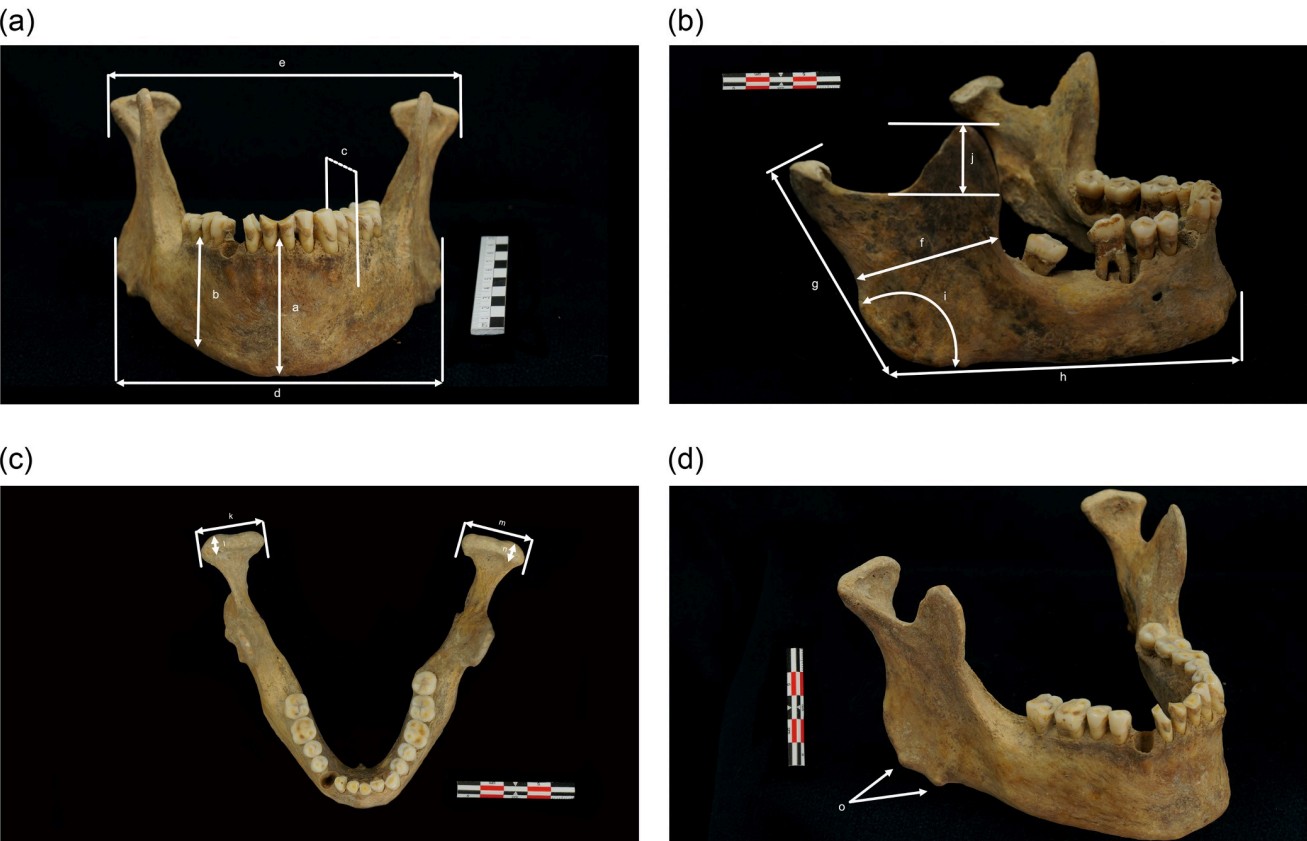

**Fig 1. a (SP 545, Les Petites Crottes collection). Measurements in the frontal plane**. (a) chin height. (b) corpus height. (c) corpus width. (d) distance between mandibular angles. (e) bicondylar width. All these measurements were taken with a sliding caliper. **b (SP 460, Ilot Saint Jacques collection). Measurements in the sagittal plane.** (f) minimum ramus width. (g) maximum ramus height. (h) corpus length. (i) mandibular angle. (j) right coronoid process height. Measurements (f) and (j) were taken with a sliding caliper. Measurements (g), (h), (i) were made with a mandibulometer. **c (SP 545, Les Petites Crottes collection). Measurements in the horizontal plane.** (k) maximum right condyle length. (l) maximum right condyle width. (m) maximum left condyle length. (n) maximum left condyle width. All these measurements were taken with a sliding caliper. **d (SP 545, Les Petites Crottes collection). Presence of exostosis at the mandibular angle.** (o) exostosis at the mandibular angle.

exhibiting taphonomic, traumatic, or pathologic changes were only included if they did not affect the measurement. The missing data rate for each parameter studied was recorded (ratio of the number of mandibles on which the parameter was measured to the total number of mandibles studied). Biological parameters (age, sex diagnosis) were also registered when the information was available.

## Statistical methods

Analyses were performed using the open-source software R software version 4.0.3, R Core Team, Vienna, Austria), available at https://www.r-project.org.

**Evaluator calibration.** To check evaluator reliability (intra-operator variability) and method reproducibility (inter-operator variability), a Bland–Altman agreement test was performed on quantitative data, and a weighted Cohen's Kappa test was run on the presence of bony exostoses at the mandibular angle, as the test was carried out on two ordinal samples.

**Sample analysis.** Mandibles from both series were pooled for statistical analysis. These were divided into two groups according to the number of exostoses at the mandibular angle:

- Group 1: total number of exostoses at the two mandibular angles, 0 or 1.

- Group 2: total number of exostoses at the two mandibular angles, greater than or equal to 2.

The F-test for the equality of two variances was used to verify normal distribution and the equality of variances between the two groups.

An analysis was carried out on the whole sample using Welch's t-test to study the correlations between the presence of exostoses at the mandibular angle and the dimensional variation of the 15 mandibular parameters investigated.

## Results

The statistical dataset is available as supplementary material (S2 Appendix).

### Sample composition

Due to the significant deterioration (e.g. mandible fracture in more than two pieces, several fragments missing) of several mandibles since the archaeological excavations, measurements could be taken on only part of the sample pre-selected from excavation reports. Thus, measurements could be made on only 50 of the 102 adult subjects in the Petites Crottes series, selected initially on the basis of the excavation report. Measurements could be made on only 30 of the 43 adult subjects in the Ilôt St Jacques series selected initially on the basis of the excavation report (S1 and S2 Tables).

Biological parameters and the number of exostoses found for the 80 subjects studied are shown in Fig 2.

### Evaluator calibration

There was a good agreement between all 1200 quantitative measurements made by the two operators (Bland–Altman tests), both in terms of repeatability for each operator (Figs 3 and 4) and inter-operator reproducibility (Fig 5).

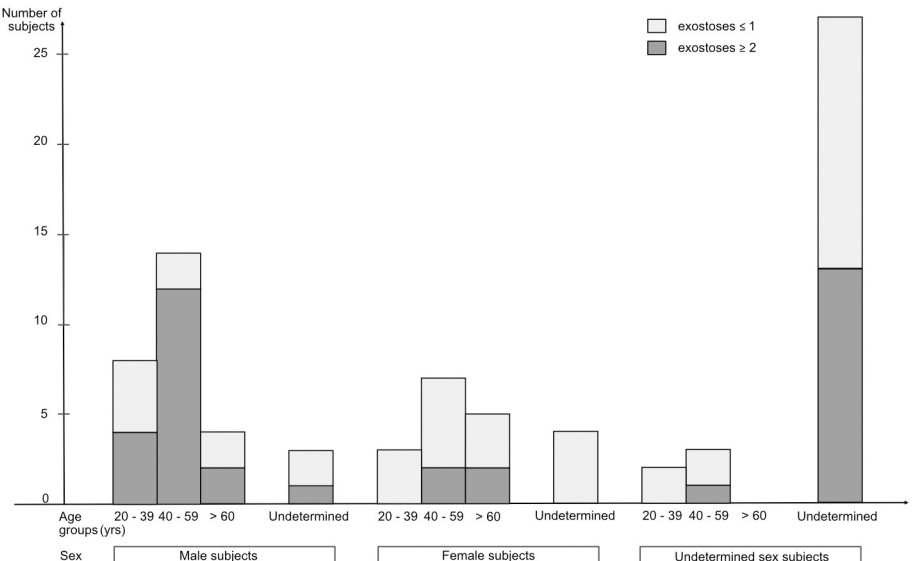

**Fig 2. Distribution of individuals according to sex, age, and presence of exostoses at the mandibular angle.** The sample comprised 29 males, 19 females, and 32 subjects of undetermined sex. Of these, 42 had a total of 0 or 1 mandibular angle exostoses for both angles (group 1), and 38 had a total of 2 or more mandibular angle exostoses for both angles (group 2).

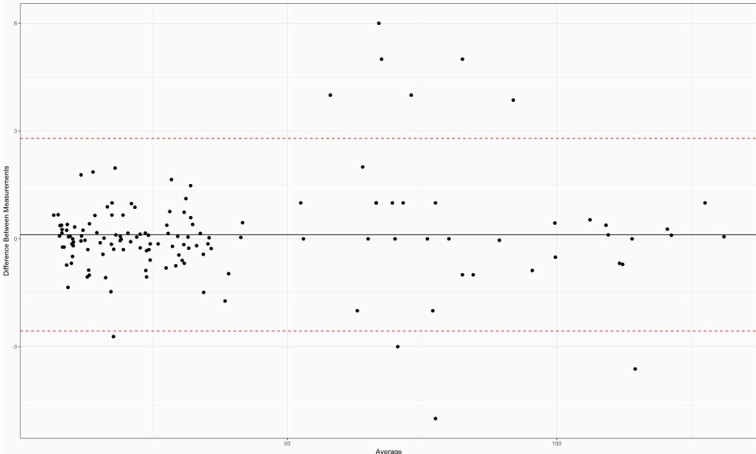

**Fig 3. Repeatability assessment for operator 1 (Bland-Altman plot).**

In addition, agreement was obtained for the count of mandibular angle exostoses with excellent reproducibility (Weighted Cohen's Kappa coefficient = 0.94 for the right side and 0.96 for the left side).

## Sample analysis

The characteristics of the 15 parameters recorded for the 80 mandibles in the sample are shown in Table 1.

The characteristics of the 15 parameters recorded for both groups are shown in Tables 2 and 3.

It thus appears that the data most frequently unable to be recorded concern the mandibular condyles and coronoid processes. These anatomical sites were most affected by taphonomic bone destruction, as they correspond to the thinnest, and therefore most fragile, zones of the mandibular bone. The conditions for the application of Welch's t-test were checked for each

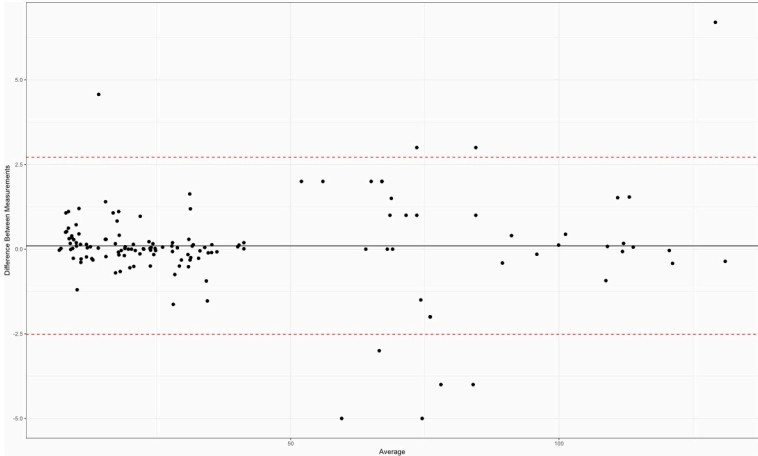

**Fig 4. Repeatability assessment for operator 2 (Bland-Altman plot).**

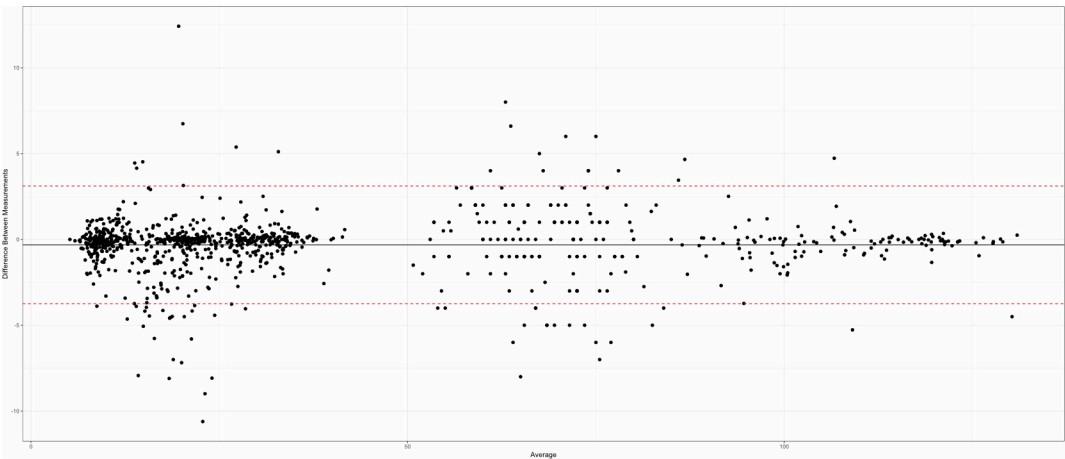

**Fig 5. Good agreement of 15 measurements for each of the 80 mandibles, between the two evaluators (Bland-Altman test).**

group using an F-test for equality of variance between the two groups, and the normality of the distribution was checked using the Q-Q plot graphical method.

The statistical analysis of the overall sample using Welch's t-test, to investigate correlations between the presence of bony exostoses at the mandibular angle and significant dimensional variation in the 15 mandibular parameters investigated, is presented in Table 4.

**Table 1. Descriptive statistics for the overall sample.**

| ROI | Minimum (mm) | Q1 (mm) | Median (mm) | Mean (mm) | Standard deviation (mm) | Q3 (mm) | Maximum (mm) | Missing data (number and percents) |
|---|---|---|---|---|---|---|---|---|
| Chin Height | 16,80 | 26,62 | 28,79 | 29,05 | 4,25 | 31,70 | 39,53 | 1 (1%) |
| Corpus Height | 14,11 | 26,86 | 28,70 | 28,61 | 4,49 | 31,62 | 38,88 | 0 (0%) |
| Corpus Width | 7,79 | 10,49 | 11,39 | 11,30 | 1,64 | 12,19 | 15,70 | 0 (0%) |
| Bigonial Breadth | 81,38 | 94,03 | 99,45 | 99,21 | 8,07 | 104,30 | 121,14 | 11 (13%) |
| Bicondylar Breadth | 102,70 | 114,50 | 119,30 | 118,80 | 6,29 | 122,20 | 130,90 | 30 (37%) |
| Minimum Ramus Width | 23,71 | 29,39 | 32,03 | 31,72 | 3,75 | 33,51 | 41,66 | 0 (0%) |
| Maximum Ramus Height | 50,75 | 58,75 | 63,00 | 62,18 | 5,33 | 65,50 | 74,25 | 1 (1%) |
| Corpus Length | 63,00 | 70,50 | 72,75 | 73,32 | 4,85 | 76,50 | 85,00 | 0 (0%) |
| Mandibular Angle■ | 107,00■ | 118,00■ | 123,20■ | 123,55■ | 7,18■ | 127,50■ | 143,50■ | 0 (0%) |
| Maximum width of Right Condyle | 5,17 | 7,78 | 8,63 | 8,67 | 1,34 | 9,40 | 13,46 | 15 (18%) |
| Maximum Length of Right Condyle | 15,36 | 19,33 | 20,64 | 20,54 | 2,10 | 22,19 | 24,35 | 20 (25%) |
| Maximum Width of Left Condyle | 5,84 | 7,87 | 8,61 | 8,68 | 1,21 | 9,46 | 12,05 | 14 (17%) |
| Maximum Length of Left Condyle | 16,60 | 19,14 | 20,64 | 20,34 | 1,76 | 21,53 | 23,73 | 16 (20%) |
| Height of Right Coronoid Process | 9,06 | 15,30 | 17,84 | 17,73 | 3,53 | 20,00 | 26,53 | 18 (22%) |
| Height of Left Coronoid Process | 9,03 | 15,33 | 17,12 | 17,37 | 3,57 | 19,80 | 26,28 | 14 (17%) |

ROI: Region Of Interest. Q1: first quartile. Q3: third quartile.

■ Mandibular angle values are expressed in degrees

**Table 2. Descriptive statistics for group 1 (42 mandibles).**

| ROI | Minimum (mm) | Q1 (mm) | Median (mm) | Mean (mm) | Standard deviation (mm) | Q3 (mm) | Maximum (mm) | Missing data (number and percents) |
|---|---|---|---|---|---|---|---|---|
| Chin Height | 16,80 | 26,62 | 28,23 | 28,45 | 4,01 | 30,62 | 36,22 | 0 (0%) |
| Corpus Height | 14,11 | 27,28 | 28,70 | 28,66 | 4,25 | 31,10 | 35,98 | 0 (0%) |
| Corpus Width | 7,79 | 10,70 | 11,34 | 11,33 | 1,76 | 12,22 | 15,70 | 0 (0%) |
| Bigonial Breadth | 81,38 | 89,29 | 94,64 | 94,88 | 7,40 | 99,45 | 113,10 | 9 (21%) |
| Bicondylar Breadth | 102,70 | 113,40 | 118,2 | 116,60 | 6,18 | 119,70 | 127,80 | 19 (45%) |
| Minimum Ramus Width | 23,71 | 28,88 | 31,61 | 31,71 | 4,16 | 33,38 | 41,66 | 0 (0%) |
| Maximum Ramus Height | 50,75 | 55,75 | 60,25 | 60,99 | 5,79 | 65,38 | 75,25 | 0 (0%) |
| Corpus Length | 63,00 | 69,50 | 72,50 | 72,68 | 5,51 | 76,25 | 85,00 | 0 (0%) |
| Mandibular Angle ■ | 107,00 ■ | 118,00 ■ | 122,50 ■ | 122,90 ■ | 6,84 ■ | 127,4 ■ | 143,50 ■ | 0 (0%) |
| Maximum width of Right Condyle | 5,17 | 7,77 | 8,41 | 8,49 | 1,48 | 9,21 | 13,46 | 8 (19%) |
| Maximum Length of Right Condyle | 16,14 | 18,84 | 20,14 | 20,21 | 2,01 | 21,16 | 24,03 | 11 (26%) |
| Maximum Width of Left Condyle | 5,84 | 7,83 | 8,61 | 8,52 | 1,20 | 9,16 | 11,02 | 9 (21%) |
| Maximum Length of Left Condyle | 17,13 | 18,17 | 19,84 | 19,65 | 1,68 | 20,96 | 23,09 | 10 (23%) |
| Height of Right Coronoid Process | 9,06 | 14,56 | 15,47 | 16,55 | 3,76 | 18,73 | 24,55 | 10 (23%) |
| Height of Left Coronoid Process | 9,03 | 13,75 | 16,70 | 16,44 | 3,79 | 18,93 | 24,39 | 7 (16%) |

ROI: Region Of Interest. Q1: first quartile. Q3: third quartile.

■ Mandibular angle values are expressed in degrees

A statistically significant difference was demonstrated between group 1 (number of exostoses ≤1) and group 2 (number of exostoses ≥ 2) for the following parameters: distance between mandibular angles, bicondylar width, ramus height, left condyle length, right and left coronoid process height. For each of these parameters, values were higher in group 2 than in group 1.

## Discussion

This cross-sectional study aimed to assess the presence of morphological changes in the mandible associated with the presence of bony exostoses at the mandibular angles. These bony prominences, because of the masseter muscle insertions they support, tend to increase in both quantity and volume when exposed to pronounced masticatory forces [21]. It was carried out on subjects recovered by archaeological excavation from two cemeteries located in the same geographical area (the city of Marseille and surrounding area, in southern France), in two series situated in the modern and contemporary historical periods.

The selection of these series was motivated by the desire to carry out a diachronic study on archaeological series presenting comparable geographical and environmental conditions, to limit the impact of extrinsic factors likely to have affected the state of health of the subjects buried in these cemeteries. In addition, all available bones were measured, so the sample size was not calculated. For this reason, all archaeological subjects who met the inclusion criteria were included in the study.

The archaeological series approach denies researchers access to information relating to the behaviours adopted by subjects during their lifetime. However, this approach allows

**Table 3. Descriptive statistics for group 2 (38 mandibles).**

| ROI | Minimum (mm) | Q1 (mm) | Median (mm) | Mean (mm) | Standard deviation (mm) | Q3 (mm) | Maximum (mm) | Missing data (number and percents) |
|---|---|---|---|---|---|---|---|---|
| Chin Height | 20,73 | 27,18 | 29,48 | 29,74 | 4,45 | 32,45 | 39,53 | 1 (2%) |
| Corpus Height | 16,45 | 25,56 | 28,66 | 28,55 | 4,79 | 31,71 | 38,88 | 0 (0%) |
| Corpus Width | 8,47 | 9,79 | 11,41 | 11,27 | 1,51 | 12,08 | 14,77 | 0 (0%) |
| Bigonial Breadth | 91,62 | 98,61 | 102,14 | 103,17 | 6,53 | 108,00 | 121,14 | 2 (5%) |
| Bicondylar Breadth | 108,80 | 117,20 | 121,00 | 120,70 | 5,83 | 124,30 | 130,90 | 11 (28%) |
| Minimum Ramus Width | 24,55 | 29,90 | 32,33 | 31,73 | 3,29 | 33,75 | 40,13 | 0 (0%) |
| Maximum Ramus Height | 54,00 | 61,00 | 63,50 | 63,52 | 4,43 | 65,50 | 74,00 | 1 (2%) |
| Corpus Length | 67,00 | 71,00 | 74,75 | 74,02 | 3,94 | 76,50 | 83,00 | 0 (0%) |
| Mandibular Angle ■ | 110,00 ■ | 118,90 ■ | 124,20 ■ | 124,30 ■ | 7,54 ■ | 128,10 ■ | 143,50 ■ | 0 (0%) |
| Maximum width of Right Condyle | 6,66 | 8,39 | 8,69 | 8,85 | 1,16 | 9,50 | 12,81 | 7 (18%) |
| Maximum Length of Right Condyle | 15,36 | 19,59 | 20,79 | 20,89 | 2,16 | 22,61 | 24,35 | 9 (23%) |
| Maximum Width of Left Condyle | 7,10 | 7,99 | 8,60 | 8,82 | 1,21 | 9,57 | 12,05 | 5 (13%) |
| Maximum Length of Left Condyle | 16,60 | 20,25 | 21,15 | 21,03 | 1,57 | 22,20 | 23,73 | 6 (15%) |
| Height of Right Coronoid Process | 15,12 | 16,80 | 18,82 | 19,00 | 2,81 | 20,18 | 26,53 | 8 (21%) |
| Height of Left Coronoid Process | 13,37 | 16,36 | 18,35 | 18,41 | 3,02 | 20,11 | 26,28 | 7 (18%) |

ROI: Region Of Interest. Q1: first quartile. Q3: third quartile.

■ Mandibular angle values are expressed in degrees

researchers to apprehend and measure distances directly, and even to evaluate volumes, freeing them from the potential bias in reconstructing cross-sections and volumes that are inherent in radiological techniques. The latter are particularly useful for assessing internal bone parameters such as bone density, as they avoid the need to destroy tissue to collect data; these two approaches therefore appear to be complementary for the study of the mandible. Data collection was closely impacted by the state of preservation of subjects and considerable variability was noted both in the number of bone pieces found during excavation and in their state of preservation in situ and then in the laboratory. Thus, while archaeological series provide access to collections of bone remains, the authors had to work with missing data, all of which represent a source of potential bias, as not all mandibles present an optimal level of preservation. Furthermore, in order not to introduce any additional bias, teeth, which are often used in the archaeological study of mandibles, were excluded from data collection, as mandibles were often found without all teeth in situ: the collection of parameters such as wear or carious lesions appeared too random to be exploitable for statistical purposes.

The 80 mandibles were divided into two groups according to the number of mandibular exostoses counted at both mandibular angles. It was decided that a subject with a unilateral exostosis associated with the absence of an exostosis at the contralateral angle would be grouped with individuals with no exostosis at either angle, as this isolated exostosis alone could not reflect overactivity of the masticatory muscles. Thus, the presence of at least two exostoses was chosen as more likely to distinguish a parafunctional increase in mandibular activity from ordinary masticatory activity [22, 23].

**Table 4. Statistical analysis of the mandibular measurements of the whole sample according to the number of exostoses.**

| ROI | Group 1 mean ±SD (mm) | Group 2 mean ±SD (mm) | Welch Two Sample t-test p-value | 95%CI |
|---|---|---|---|---|
| Chin Height | 28.45±4.01 | 29.74±4.46 | 0,183 | -3.199 0.623 |
| Corpus Height | 28.66±4.25 | 28.55±4.79 | 0,911 | -1.915 2.142 |
| Corpus Width | 11.33±1.76 | 11.27±1.51 | 0,876 | -0.671 0.785 |
| Bigonial Breadth | 94.88±7.40 | 103.17±6.53 | 0,000006513** | -11.657–4.917 |
| Bicondylar Breadth | 116.60±6.19 | 120.70±5.84 | 0,02051* | -7.549–0.662 |
| Minimum Ramus Width | 31.71±4.16 | 31.73±3.29 | 0,981 | -1.682 1.643 |
| Maximum Ramus Height | 60.99±5.79 | 63.52±4.44 | 0,03103* | -4.834–0.237 |
| Corpus Length | 72.68±5.51 | 74.02±3.94 | 0,2138 | -3.453 0.785 |
| MandibularAngle■ | 122.87±6.85■ | 124.31±7.55■ | 0,3761 | -4.662 1.781 |
| Maximum width of Right Condyle | 8.49±1.48 | 8.86±1.17 | 0,271 | -1.023 0.292 |
| Maximum Length of Right Condyle | 20.21±2.02 | 20.89±2.16 | 0,2124 | -1.765 0.401 |
| Maximum Width of Left Condyle | 8.53±1.20 | 8.83±1.21 | 0,317 | -0.892 0.294 |
| Maximum Length of Left Condyle | 19.65±1.68 | 21.03±1.58 | 0,00115* | -2.202–0.574 |
| Height of Right Coronoid Process | 16.55±3.76 | 19.00±2.82 | 0,005026* | -4.136–0.769 |
| Height of Left Coronoid Process | 16.44±3.79 | 18.41±3.03 | 0,02237* | -3.650–0.288 |

ROI: Region Of Interest. SD: Standard Deviation. 95%CI: 95% confidence interval. p: significance level;

*significance level, $p < 0.05$;

**significance level, $p < 0.001$

■ Mandibular angle values are expressed in degrees

The mandibular characteristics associated with the presence of exostoses at the mandibular angle have been studied in the literature. Indeed, several studies [9, 22, 24] associate the presence of mandibular exostoses with the expression of a parafunctional activity called bruxism, which reflects the intense activity of the muscles of the masticatory apparatus. Other articles focus on the development of mandibular tori, which are oral exostoses located on the medial aspect of the mandibular arch. Research has also shown that signs of parafunctional activity are associated with the presence of mandibular tori [25].

The impact on the mandible of these parafunctional muscular activities associated with bruxism has been the subject of various published studies, based mainly on two-dimensional medical imaging, and involving the analysis of panoramic radiographs. Morphological changes in the mandible have been described, which sometimes contradict the data collected in this study. As a reminder, the study of archaeological collections has highlighted several morphological changes in the mandible, relating to parameters such as the distance between the mandibular angles, bicondylar width, ramus height, left condyle length, and right and left coronoid process height. The values of all these parameters were increased in subjects with multiple exostoses at the mandibular angles. These changes affected several anatomical regions of the mandible.

Various results have been presented for the condylar region of the mandible: the present study showed an increase in the length of the left mandibular condyle; one study [26] found no significant difference; several others [27–29] showed a reduction in condylar dimensions. This result should be treated with caution, as it was demonstrated for the left condyle only. Measurements of right condyle length, as well as maximum condyle widths on both sides, showed no statistically significant difference between the two groups. Furthermore, many factors have been described in the literature as inducing variations in mandibular

condyle shape, such as age, gender, facial typology, and intermaxillary dental relationships [30, 31]. This ability to remodel the shape of the condyle reflects its functional adaptation throughout an individual's life, and the resulting modifications are often described as asymmetrical.

A bilateral increase in dimensions was found in the coronoid process region in this study. This is corroborated by the study of Cezairli *and al.* [32]; Padmaja Satheeswarakumar *et al.* [27] observed coronoid process measurements in bruxers that were significantly smaller on the right but larger on the left. A possible explanation for this size increase could relate to the insertion of the temporalis muscle and the deep fascicle of the masseter muscle on the lateral aspect of the coronoid processes: sustained muscular activity could explain the higher level of bone development found in patients with multiple exostoses.

Finally, this study reported an increased distance between the mandibular angles and between the lateral poles of the condyles in subjects with multiple exostoses compared to the other group. Although these measurements were not collected in the medical imaging studies, it is possible to advance an explanatory hypothesis based on knowledge of mandibular biomechanics. Indeed, Van Eijiden [33] reported that during the contraction of the elevator muscles, the mandible undergoes lateral flexion in the transverse direction, resulting in lateral torsion of the lower mandibular border. It thus seems possible to suggest that repetition of these muscle contractions over time has resulted in a widening of the intercondylar distance, and consequently an increased intercondylar distance due to the anatomical configuration of the mandible.

Thus, eighty adult human mandibles from two archaeological series were included in this study and divided into two groups according to the number of bony exostoses at the mandibular angle. The sixteen measurements taken on each mandible concerned only the external characteristics of the bones (length, width, thickness, angulation). A statistically significant difference between group 1 (number of exostoses $\leq 1$) and group 2 (number of exostoses $\geq 2$) for the following parameters: distance between mandibular angles, bicondylar width, ramus height, left condyle length, right and left coronoid process height. For each parameter, values were higher in group 2 than in group 1.

## Conclusion

This study presents an original methodology for studying anatomical variations of the mandible in the context of parafunctional activity, highlighting certain mandibular modifications. The presence of bilateral mandibular exostoses was correlated with higher values for the distance between mandibular angles, bicondylar width, ramus height, left condylar length, and right and left coronoid process height. It would be interesting to apply the methodological approach proposed in this study to other archaeological series, different in terms of location and period of life of the subjects. The variations observed corroborated the results of other studies based on medical imaging, obtained from medical imaging. The use of imaging examinations could represent a potentially valuable complementary research approach, whether applied to archaeological series or patient cohorts and would appear to be a sensible option for future research, both in terms of bone density and morphological variations.

The impact of parafunctional behaviours such as bruxism on the mandible therefore has many anatomical expressions. The considerable variability of results found in the literature shows that more studies are needed to reach a consensus on the impact of parafunctional activities on the mandible.

## Supporting information

**S1 Appendix. STROBE checklist.**
(PDF)

**S2 Appendix. Statistical dataset.**
(RAR)

**S1 Table. Measurement table for operator 1.**
(CSV)

**S2 Table. Measurement table for operator 2.**
(CSV)

## Author Contributions

**Conceptualization:** Estelle Casazza.

**Formal analysis:** Benoit Ballester.

**Investigation:** Estelle Casazza, Benoit Ballester.

**Methodology:** Estelle Casazza, Anne Raskin.

**Project administration:** Yann Ardagna.

**Resources:** Yann Ardagna.

**Supervision:** Anne Raskin.

**Validation:** Anne Raskin.

**Visualization:** Camille Philip-Alliez.

**Writing – original draft:** Estelle Casazza.

**Writing – review & editing:** Estelle Casazza.

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
