## [Decision Letter · Decision Letter 0]

13 Aug 2024

PONE-D-24-24371Morphological changes in the mandible associated with the presence of exostoses: a study in two archaeological populations from southern France.PLOS ONE

Dear Dr. CASAZZA,

Thank you for submitting your manuscript to PLOS ONE. After careful consideration, we feel that it has merit but does not fully meet PLOS ONE’s publication criteria as it currently stands. Therefore, we invite you to submit a revised version of the manuscript that addresses the points raised during the review process.

We look forward to receiving your revised manuscript.

Kind regards,

Sameh Attia, MS

Academic Editor

PLOS ONE

2. "In your manuscript, please provide additional information regarding the specimens used in your study. Ensure that you have reported human remain specimen numbers and complete repository information, including museum name and geographic location. 

For more information on PLOS ONE's requirements for paleontology and archeology research, see https://journals.plos.org/plosone/s/submission-guidelines#loc-paleontology-and-archaeology-research.

4. We note that Figure 1,a,b,c,d in your submission contain [map/satellite] images which may be copyrighted. All PLOS content is published under the Creative Commons Attribution License (CC BY 4.0), which means that the manuscript, images, and Supporting Information files will be freely available online, and any third party is permitted to access, download, copy, distribute, and use these materials in any way, even commercially, with proper attribution. For these reasons, we cannot publish previously copyrighted maps or satellite images created using proprietary data, such as Google software (Google Maps, Street View, and Earth). For more information, see our copyright guidelines: http://journals.plos.org/plosone/s/licenses-and-copyright.

1. You may seek permission from the original copyright holder of Figure 1,a,b,c,d to publish the content specifically under the CC BY 4.0 license.  

Additional Editor Comments (if provided):

Reviewers' comments:

Reviewer's Responses to Questions

**Comments to the Author**

1. Is the manuscript technically sound, and do the data support the conclusions?

Reviewer #1: Partly

Reviewer #2: Yes

Reviewer #3: Partly

2. Has the statistical analysis been performed appropriately and rigorously? 

Reviewer #1: Yes

Reviewer #2: Yes

Reviewer #3: No

3. Have the authors made all data underlying the findings in their manuscript fully available?

Reviewer #1: Yes

Reviewer #2: Yes

Reviewer #3: No

4. Is the manuscript presented in an intelligible fashion and written in standard English?

Reviewer #1: Yes

Reviewer #2: Yes

Reviewer #3: No

5. Review Comments to the Author

Reviewer #1: The work presented for evaluation is an important addition to the knowledge regarding the morphological characteristics of human mandibles from archaeological materials in relation to exostoses. The work arrangement is appropriate, the goal has been clearly defined and the results are presented transparently. I have a minor comment regarding the title. It should be supplemented with information that the material consisted of human mandibles. This also applies to the content of the article. The conclusions are quite incomprehensible to me. In lines 305-306 they state: "we did not investigate certain characteristics, such as bone density, by observation of trabecular bone, as study of the collections required the use of non-invasive methods". Firstly, bone densitometry is a non-invasive method, and secondly, it does not examine the trabecular structure of the bone. L of line 308 we read "Mandibular bone density has been analyzed on panoramic X-rays.." I found no mention of this in the methodology and results. Was this a discussion with other authors? I don't see any quote.

Reviewer #2: hello

thank you for an interesting paper

from clinical and anatomical point of view there is no manducation but mastication - please change in the abstract and introduction to: - mastication (manducation) apparatus in the abstract and chapter introduction. in chapter introduction write the difference between manducation and mastication and later use the mastication term

probably the definition is based on some cultural, anatomical and medical term differences, like:

look here- https://link.springer.com/chapter/10.1007/978-3-030-13739-7_21

since the mandibular angle has its protuberances, the muscular fibers of the masseter muscle tends to attach to those bony parts, therefore the mastication is possible

example of paper in french: - mascication muscles - https://pubmed.ncbi.nlm.nih.gov/31626502/

its a great differences in the muscle nomenclature based on european, english and american boards of anatomy

in the introduction add the muscles of mastication and the role of masseter muscle, which function is greately dependable on chewing and biting;

the function of mastication or the function of manducation?

an important remark should be addresed in the mandibular angles anatomy, shape and build, In some cases of severe mastication forces, bite and muscles overgrowth the mandibular angles tend to rotate, angulate or bow themselves because of the extensive forces.

introduction is sufficient

mate4rial and methods are described

please write the full meaning of SRA DRAC-PACA + add the number/year of approval form for the study

inclusion and exclusion criteria for study are described

what does a breadth distance mean?

what calipers were used? company/name/country of original data is missing - improve a detailed information about the used devices for the study

what is the R-software? - same: add more details bout this software, producent, version, country of origins

its important to explain in detailed used material and methods so that any reader can used the same devices/methods in the future

sample analysis is ok

results section: - line 180-184 - why only some skulls were sufficient for measurements?> what factors did influence on this situation - describe them

the two operators data should be blind-folded, like for example - examiner 1 (X.X), and examiner 2 (Y.Y)

was it possible to distinguish male/female skulls? does gender/age associate with the tuberances count? was this possible to establish or is it the study limitation?

results are sufficient

results chapter is well written, im just missing a 1-2 sentence comment on each table in the result section - please add them

discussion section:

-first paragraph in the discussion section should be improved and more information about mandibular angle changes in shape, size, build and protuberance occurence should be added

discussion is propper, however im missing top 5 key highlights from author study at the end of this discussion section

the final conclusions - should be re-arranged and re-wrriten since uthors didnt use any rtg/cbct studies just some calipers - please change and improve the final conclusions

referenes are sound

presented figures are sufficient - im missing some more explanation remakrs belowe each figure description

same the used calipers, distances and measuring reference points should be more careuflyl described and presented

Reviewer #3: Although this study is interesting and valuable, the paper is not good and has some important issues to resolve.

The important points to be corrected are:

1. The manuscript is written very poorly. Many points remain for the reader to guess (or get confused over them). Some necessary information that should have been reported in the Methods section are reported in the Results section. Many other important Methodological details are completely missing. Overall, the paper is written poorly and should be completely re-written at the Methods and Results part.

2. For starters, please apply the STROBE checklist's items to the revision.

3. Fill out the STROBE checklist with page numbers and submit it too.

4. The abstract is as well written very poorly. Overall, it seems that the paper is written mostly by inexperienced author(s). The current authors are not qualified to handle the revisions. They should definitely seek help from a senior, experienced professor in writing the paper.

5. The discussion should discuss the lack of sample size calculations before conducting the study.

Before any in-depth reviews, I need the paper to be re-written into a fluent, organized, and correct format. Then I will be able to judge its scientific merit and scientific points.

6. PLOS authors have the option to publish the peer review history of their article (what does this mean?). If published, this will include your full peer review and any attached files.

Reviewer #1: No

Reviewer #2: No

Reviewer #3: No

---

## [Author Response · Author response to Decision Letter 0]

14 Sep 2024

Dear Reviewers,

Thank you for your careful review of our manuscript.

We have considered every one of your comments to improve this article. You will find the answers following each request.

Kind regards,

The authors

Reviewer #1: 

The work presented for evaluation is an important addition to the knowledge regarding the morphological characteristics of human mandibles from archaeological materials in relation to exostoses. The work arrangement is appropriate, the goal has been clearly defined and the results are presented transparently. I have a minor comment regarding the title. It should be supplemented with information that the material consisted of human mandibles. This also applies to the content of the article.

=> We want to thank you for your pertinent comments and your sound advice. We've changed that (lines 1, 20, 79, 339)

The conclusions are quite incomprehensible to me. In lines 305-306 they state: "we did not investigate certain characteristics, such as bone density, by observation of trabecular bone, as study of the collections required the use of non-invasive methods". Firstly, bone densitometry is a non-invasive method, and secondly, it does not examine the trabecular structure of the bone. L of line 308 we read "Mandibular bone density has been analyzed on panoramic X-rays.." I found no mention of this in the methodology and results. Was this a discussion with other authors? I don't see any quote.

=> We have taken all your suggestions into account to complete the manuscript. We hope that this new submission will respond fully to your feedback (lines 348 – 366)

Reviewer #2: 

Hello thank you for an interesting paper

from clinical and anatomical point of view there is no manducation but mastication - please change in the abstract and introduction to: - mastication (manducation) apparatus in the abstract and chapter introduction. in chapter introduction write the difference between manducation and mastication and later use the mastication term

probably the definition is based on some cultural, anatomical and medical term differences, like:

look here- https://link.springer.com/chapter/10.1007/978-3-030-13739-7_21

=>Thank you for your relevant comment. Indeed, the term “manducation” is rarely used in international literature, and it is mainly used by French authors. It has therefore been removed from this manuscript to avoid misunderstandings (lines 23, 43, 69, 76, 77, 296, 301).

since the mandibular angle has its protuberances, the muscular fibers of the masseter muscle tends to attach to those bony parts, therefore the mastication is possible

example of paper in french: - mascication muscles - https://pubmed.ncbi.nlm.nih.gov/31626502/

=>The introduction has been completed according to your suggestions (lines 67–73).

its a great differences in the muscle nomenclature based on european, english and american boards of anatomy

in the introduction add the muscles of mastication and the role of masseter muscle, which function is greately dependable on chewing and biting; 

=>Thank you for your comment. It has been changed (lines 46–48).

the function of mastication or the function of manducation? Masticatory function

an important remark should be addresed in the mandibular angles anatomy, shape and build, In some cases of severe mastication forces, bite and muscles overgrowth the mandibular angles tend to rotate, angulate or bow themselves because of the extensive forces. 

=>Thank you for your comment. It has been changed (lines72-73).

introduction is sufficient

mate4rial and methods are described

please write the full meaning of SRA DRAC-PACA + add the number/year of approval form for the study : Thank you for your comment. 

=>It has been changed (lines 85-87).

inclusion and exclusion criteria for study are described

what does a breadth distance mean? The measure of the distance from side to side of a solid object

what calipers were used? company/name/country of original data is missing - improve a detailed information about the used devices for the study : 

=>Thank you for your comment. It has been changed (line 177, lines 131–156).

what is the R-software? - same: add more details bout this software, producent, version, country of origins 

=> R is a free software environment for statistical computing and graphics. Some details have been added in the manuscrit (lines 187-188).

its important to explain in detailed used material and methods so that any reader can used the same devices/methods in the future

sample analysis is ok

results section: - line 180-184 - why only some skulls were sufficient for measurements?> what factors did influence on this situation - describe them 

=> It has been done (lines 207 - 209)

the two operators data should be blind-folded, like for example - examiner 1 (X.X), and examiner 2 (Y.Y) (line 175)

was it possible to distinguish male/female skulls? does gender/age associate with the tuberances count? was this possible to establish or is it the study limitation? 

=>Thank you for your pertinent comment. Indeed this is a limitation of the study, because more than a third of the sample (32 subjects/80) could not benefit from a sex diagnosis, and an equivalent proportion did not have an estimated age: we did not have enough individuals to be able to explore these two parameters (lines 184-185).

results are sufficient

results chapter is well written, im just missing a 1-2 sentence comment on each table in the result section - please add them Thank you for your comment. 

=> It has been changed (lines 242–244, lines 258–261).

discussion section:

-first paragraph in the discussion section should be improved and more information about mandibular angle changes in shape, size, build and protuberance occurence should be added 

=> Thank you for your comment. It has been changed (lines 264–266).

discussion is propper, however im missing top 5 key highlights from author study at the end of this discussion section

=> Thank you for your advice, this part of the discussion has been added (lines 339–345).

the final conclusions - should be re-arranged and re-wrriten since uthors didnt use any rtg/cbct studies just some calipers - please change and improve the final conclusions referenes are sound presented figures are sufficient 

=> It has been done (lines 352-363)

- im missing some more explanation remakrs belowe each figure description same the used calipers, distances and measuring reference points should be more careuflyl described and presented

=> Your comment has been taken into account and the measurements have been more precisely described (lines 131-156, 162, 166, 170).

Reviewer #3: 

Thank you for the attention you have given to reading this manuscript.

We have taken your comments into account and have tried to make the necessary changes to improve the manuscript and the editorial quality of the article. We hope that these changes will meet your expectations.

Although this study is interesting and valuable, the paper is not good and has some important issues to resolve.

The important points to be corrected are:

1. The manuscript is written very poorly. Many points remain for the reader to guess (or get confused over them). Some necessary information that should have been reported in the Methods section are reported in the Results section. Many other important Methodological details are completely missing. Overall, the paper is written poorly and should be completely re-written at the Methods and Results part. 

=>We have taken your comments into account, and have revised the Method and Results sections.

2. For starters, please apply the STROBE checklist's items to the revision. 

=>Thank you for this advice. We have carefully followed the items in the STROBE checklist to review this work.

3. Fill out the STROBE checklist with page numbers and submit it too. 

=> Thank you for your comment. We've changed this point. STROBE checklist has been submitted too.

4. The abstract is as well written very poorly. Overall, it seems that the paper is written mostly by inexperienced author(s). The current authors are not qualified to handle the revisions. They should definitely seek help from a senior, experienced professor in writing the paper. 

=> We have carried out the revisions according to your advice.

5. The discussion should discuss the lack of sample size calculations before conducting the study. 

=> We would like to thank you for your helpful comments (lines 273-274)

Before any in-depth reviews, I need the paper to be re-written into a fluent, organized, and correct format. Then I will be able to judge its scientific merit and scientific points.

---

## [Decision Letter · Decision Letter 1]

8 Oct 2024

PONE-D-24-24371R1Morphological changes in the human mandible associated with the presence of exostoses: a cross-sectional study in two archaeological populations from southern France.PLOS ONE

Dear Dr. CASAZZA,

Thank you for submitting your manuscript to PLOS ONE. After careful consideration, we feel that it has merit but does not fully meet PLOS ONE’s publication criteria as it currently stands. Therefore, we invite you to submit a revised version of the manuscript that addresses the points raised during the review process.

We look forward to receiving your revised manuscript.

Kind regards,

Sameh Attia, MS

Academic Editor

PLOS ONE

Reviewers' comments:

Reviewer's Responses to Questions

**Comments to the Author**

1. If the authors have adequately addressed your comments raised in a previous round of review and you feel that this manuscript is now acceptable for publication, you may indicate that here to bypass the “Comments to the Author” section, enter your conflict of interest statement in the “Confidential to Editor” section, and submit your "Accept" recommendation.

Reviewer #1: All comments have been addressed

Reviewer #2: All comments have been addressed

Reviewer #3: (No Response)

2. Is the manuscript technically sound, and do the data support the conclusions?

Reviewer #1: Yes

Reviewer #2: Yes

Reviewer #3: Yes

3. Has the statistical analysis been performed appropriately and rigorously? 

Reviewer #1: Yes

Reviewer #2: Yes

Reviewer #3: Yes

4. Have the authors made all data underlying the findings in their manuscript fully available?

Reviewer #1: Yes

Reviewer #2: Yes

Reviewer #3: No

5. Is the manuscript presented in an intelligible fashion and written in standard English?

Reviewer #1: Yes

Reviewer #2: Yes

Reviewer #3: Yes

6. Review Comments to the Author

Reviewer #1: (No Response)

Reviewer #2: Thank you for the review

paper is improved

all the necessary changes are done

figures and tables meet the paper criteria

thank you

Reviewer #3: The revision is better but there is a lot of room for improvement. Please improve it more:

- While reporting p-values, please report at least up to 3 decimal places or more, but not fewer.

- Besides reporting min, max, mean, and SD, also do report median, Q1, and Q3.

- Also report 95% Confidence Intervals.

- Report these parameters for the whole sample as well as both groups.

- Attach the statistical raw dataset as supplementary material.

- The paper is still very unorganized. The organization of the paper is done very poorly. The subheadings are the same size as the main headings. You should use different levels of subheadings. Avoid using the font for the main headings for the subheadings.

- Add numbers to the headings and subheadings for better fluency and organization. Like 3, 3.1, 2.3.1...

- The name and details of the statistical software?

7. PLOS authors have the option to publish the peer review history of their article (what does this mean?). If published, this will include your full peer review and any attached files.

Reviewer #1: No

Reviewer #2: No

Reviewer #3: No

---

## [Author Response · Author response to Decision Letter 1]

12 Oct 2024

Dear Editor, dear Reviewers,

Thank you for your careful review of our manuscript.

We have considered every one of your comments to improve this article. You will find our responses to each of your suggestions in blue.

Kind regards,

The authors

Reviewer #1: (No Response)

Reviewer #2:

Thank you for the review

paper is improved

all the necessary changes are done

figures and tables meet the paper criteria

thank you

We want to thank you once again for your careful proofreading and the valuable advice you gave us to improve the manuscript.

Reviewer #3: The revision is better but there is a lot of room for improvement. Please improve it more:

- While reporting p-values, please report at least up to 3 decimal places or more, but not fewer.

Thank you for your relevant comment. We've changed that (line 259).

- Besides reporting min, max, mean, and SD, also do report median, Q1, and Q3.

We've added that (line 235).

- Also report 95% Confidence Intervals.

We've added that (line 259).

- Report these parameters for the whole sample as well as both groups.

Thank you for your relevant comment. We've added two tables (lines 240 and 243)

- Attach the statistical raw dataset as supplementary material. 

Thank you for your suggestion. The statistical raw dataset has been attached as supplementary material (Appendix S2).

- The paper is still very unorganized. The organization of the paper is done very poorly. The subheadings are the same size as the main headings. You should use different levels of subheadings. Avoid using the font for the main headings for the subheadings.

- Add numbers to the headings and subheadings for better fluency and organization. Like 3, 3.1, 2.3.1...

Thank you for your advice. However, the PLOS ONE manuscript body formatting guidelines, which we have carefully followed, does not mention the addition of numbers. The choice different levels of subheadings has also been calibrated according to PLOS ONE instructions. For these reasons, we have not been able to modify the manuscript according to your suggestions.

- The name and details of the statistical software?

These details had been given lines 184-185.

---

## [Decision Letter · Decision Letter 2]

19 Nov 2024

Morphological changes in the human mandible associated with the presence of exostoses: a cross-sectional study in two archaeological populations from southern France.

PONE-D-24-24371R2

Dear Dr. CASAZZA,

We’re pleased to inform you that your manuscript has been judged scientifically suitable for publication and will be formally accepted for publication once it meets all outstanding technical requirements.

Kind regards,

Sameh Attia, MS

Academic Editor

PLOS ONE

Reviewers' comments:

Reviewer's Responses to Questions

**Comments to the Author**

1. If the authors have adequately addressed your comments raised in a previous round of review and you feel that this manuscript is now acceptable for publication, you may indicate that here to bypass the “Comments to the Author” section, enter your conflict of interest statement in the “Confidential to Editor” section, and submit your "Accept" recommendation.

Reviewer #3: (No Response)

2. Is the manuscript technically sound, and do the data support the conclusions?

Reviewer #3: (No Response)

3. Has the statistical analysis been performed appropriately and rigorously? 

Reviewer #3: (No Response)

4. Have the authors made all data underlying the findings in their manuscript fully available?

Reviewer #3: (No Response)

5. Is the manuscript presented in an intelligible fashion and written in standard English?

Reviewer #3: (No Response)

6. Review Comments to the Author

Reviewer #3: The revision is satisfactory. I think the manuscript is acceptable for publication.

Congratulations.

7. PLOS authors have the option to publish the peer review history of their article (what does this mean?). If published, this will include your full peer review and any attached files.

Reviewer #3: No

---

## [Editor Report · Acceptance letter]

22 Nov 2024

PONE-D-24-24371R2 

PLOS ONE

Dear Dr. CASAZZA, 

I'm pleased to inform you that your manuscript has been deemed suitable for publication in PLOS ONE. Congratulations! Your manuscript is now being handed over to our production team.

Kind regards, 

on behalf of

Dr. Sameh Attia 

Academic Editor

PLOS ONE